# The Effect of Fresh Kale (*Brassica oleracea* var. *sabellica*) Addition and Processing Conditions on Selected Biological, Physical, and Chemical Properties of Extruded Snack Pellets

**DOI:** 10.3390/molecules28041835

**Published:** 2023-02-15

**Authors:** Jakub Soja, Maciej Combrzyński, Tomasz Oniszczuk, Beata Biernacka, Agnieszka Wójtowicz, Karol Kupryaniuk, Karolina Wojtunik-Kulesza, Maciej Bąkowski, Marek Gancarz, Jarosław Mołdoch, Jarosław Szponar, Anna Oniszczuk

**Affiliations:** 1Department of Thermal Technology and Food Process Engineering, University of Life Sciences in Lublin, Głęboka 31, 20-612 Lublin, Poland; 2Department of Inorganic Chemistry, Medical University of Lublin, Chodźki 4a, 20-093 Lublin, Poland; 3Institute of Animal Nutrition and Bromatology, University of Life Sciences in Lublin, Akademicka 13, 20-950 Lublin, Poland; 4Faculty of Production and Power Engineering, University of Agriculture in Krakow, Balicka 116b, 30-149 Krakow, Poland; 5Institute of Agrophysics Polish Academy of Sciences, Doświadczalna 4, 20-290 Lublin, Poland; 6Department of Biochemistry and Crop Quality, Institute of Soil Science and Plant Cultivation, State Research Institute, 24-100 Puławy, Poland; 7Toxicology Clinic, Clinical Department of Toxicology and Cardiology, Medical University of Lublin, Stefan Wyszyński Regional Specialist Hospital, Al. Kraśnicka 100, 20-718 Lublin, Poland

**Keywords:** extrusion-cooking, fresh kale, snack pellets, antioxidant activity, biological and physicochemical properties

## Abstract

The purpose of this study was to determine the effect of the addition of fresh kale and processing conditions on extruded pellet antioxidant activity and selected physicochemical properties. The results of the applied DPPH, FRAP, and TPC methods indicated that, for both 60 and 100 rpm screw speeds, snack pellet antioxidant activity and phenolic content were strongly linked to the fresh kale content, and these properties increased with the addition of this plant. The amount of fresh kale and the applied processing variables (extruder screw speed and the moisture content of the raw material blends) were also found to significantly affect the water absorption index, water solubility index, fat absorption index, fatty acid profile, and basic chemical composition of the obtained extrudates. The sample with the highest phenolic content (72.8 μg GAE/g d.w.), the most advantageous chemical composition (protein, ash, fat, carbohydrates, and fiber content), and high antioxidant properties was produced at a fresh kale content of 30%, a 36% moisture content, and a 100 rpm screw speed. The following phenolic acids were identified in this sample: protocatechuic, 4-OH-benzoic, vanillic, syringic, salicylic, caffeic, coumaric, ferulic, and sinapic. Sinapic acid was the prevailing phenolic acid.

## 1. Introduction

The global market for food and food technology is developing rapidly, and investment in this sector is crucial in many international development plans. High-quality food intake is a basic need for living, growing, surviving, and maintaining proper functioning of the body. The growing demand for food encourages both producers and consumers to look for alternative sources of food with high nutritional value [1]. Nowadays, it is extremely important that scientists use an interdisciplinary approach in food research. This allows for changes in the design and management of food systems in order to balance the use of natural resources, while taking into account the needs of a constantly developing society. It should be noted that food production significantly affects the natural world. There is, therefore, an urgent need to move to long-term sustainable food resource development due to the constantly emerging problems of natural resource depletion, the degradation of the environment, the reduction of biodiversity, constant hunger in the world, climate change, and the decreasing area of agricultural lands [2,3].

Currently, the market for extruded snacks is mainly based on products obtained from snack pellets after additional final expansion. They are obtained via extrusion-cooking, which is a pressure-thermal treatment (HTST, i.e., high temperature short time), during which the raw materials are processed in a device called an ‘extruder’ under high temperature and pressure and the presence of shear forces [4]. The basic raw materials for such items are potato products (starch, flakes, or grits), wheat or corn flour or starch, and additives affecting physicochemical and sensory properties, i.e., oil, sugar, and salt. Today, there are several types of extruded snacks available on the market, including first, second, and third generation snacks, in which appearance and structure depend primarily on the type of matrix and the plasticizing system [5].

Consumers are increasingly paying attention to the composition and health-promoting potential of food products and are looking for items that stand out from others in the same product group. This is due to the increase in nutrition awareness among consumers, who now demand complete and wholesome products that are able to replace permanent meals or diversify their diet. Properly balanced products allow the provision of valuable nutrients, thus classifying them into a group of products called ‘healthy foods’ [6].

Fruits or vegetables are often added to the mixture in order to enrich the nutritional value of snacks. Their addition greatly affects snack pellet properties and significantly increases the attractiveness of the product. Extruded snacks enriched with vegetable or fruit additives have more protein, fiber, vitamins, and antioxidants. Such action allows for the creation of high-quality products of a functional food character, which, in the era of increased consumer nutrition awareness, is desirable by many social groups who are following healthy lifestyles [7].

Kale (*Brassica oleracea* var. *sabellica*) is a leafy plant that, depending on the variety, has a frayed leaf structure with a green or red-violet color. It is characterized by a high content of glucosinolates and has significant amounts of phenolic compounds, including quertecin and kaempferol. Research conducted by Korus et al. [8] showed that the content of polyphenols in 100 g of fresh vegetables was in the range of 256–531 mg. Kale is a vegetable with a very high concentration of vitamins C, A, B_1_, B_2_, B_6_, and E [9]. Moreover, it contains dietary fiber, micronutrients (i.e., iron, zinc, and manganese) and macroelements (i.e., calcium and magnesium) [10]. Much research shows that the consumption of the phenolic compounds contained in fresh kale has a positive impact on the prevention of chronic cardiovascular diseases [10,11,12] and contributes to the reduction of the risk of coronary artery disease [10]. In addition, fresh kale is a source of lutein (about 40 mg/100 g of fresh product), which prevents eye damage induced by reactive oxygen species [12]. Furthermore, its addition gives the products a desirable taste, flavor, and appearance; thus, products enriched with this type of vegetable stand out from other products of the same group. As a result, in recent years, this vegetable has gained great popularity in the food industry and is often referred to as a ‘superfood’ [13].

The aim of this study was to examine how the addition of fresh kale to extruded snack pellets affects their selected physicochemical properties under various processing conditions and how the products obtained stand out from other extrudates in terms of chemical composition and antioxidant properties.

## 2. Results and Discussion

### 2.1. Antioxidant Activity and Phenolic Compound Content of Snack Pellets Enriched with Fresh Kale

The wide spectrum of health-promoting activities obtained from fresh kale and the presence of chemical constituents with high therapeutic possibilities make this plant a candidate for the development of functional food products. The samples, snack pellets with different quantities of fresh kale (0, 10, and 30%), were produced via extrusion-cooking at various initial moisture contents and at various screw speeds. In the first phase of the experiment, the authors examined the sample antioxidant potential.

DPPH enabled the determination of the extrudate extract free radical scavenging ability. The results explicitly indicated high antioxidant activity within all extracts. Moreover, activity increased significantly with the amount of added fresh kale. In the case of the 30% content of fresh kale, the addition displayed 10% and 27% higher free radical scavenging ability (depending on processing conditions) after the same time and in comparison to the 10% fresh kale content, respectively.

The obtained results were also analyzed using the Trolox equivalent antioxidant activity (TEAC). The TEAC value was calculated based on a standard curve wherein y = −0.0214x + 0.471. Table 1 presents the TEAC values of activity at 10 min from the reaction initiation.

In all cases, fresh kale enrichment positively influenced the free radical scavenging of the analyzed extrudate extracts, in comparison to extracts of potato basic blends. Herein, an increase in antioxidant activity was observable in the first minutes of analysis, and by the end of the experiment, the increase was equal to 37.19% for the 30% fresh kale addition and 11.39% for the 10% fresh kale addition.

Initial moisture content (mc) and screw rotation speed (rpm) also had significant impacts on the scavenging activity against DPPH. We observed that higher parameters (mc = 36%, rpm = 100) were more effective in enhancing scavenging activity for higher fresh kale content (30%), whereas 32% mc and 60 rpm were more suitable for lower fresh kale content (10%). 

In the next step of the experiment, FRAP spectrophotometric analysis was applied to determine the ability of the samples to reduce iron (III) ions. We assumed that the higher the FRAP value in the tested substance, the greater its reducing power. The obtained results are shown in Table 2.

The converted results for both FRAP units and gallic acid equivalents indicated that the reducing activity is highly linked to the fresh kale content, and for both 60 and 100 rpm processing speeds, it increased with the addition of this plant (Table 2).

The most valuable parameters were obtained with the 30% fresh kale addition, whereas 10% revealed only a slight increase in Fe^3+^ reduction ability in comparison to potato control samples. Similarly to other study results, the FRAP method revealed significant dependencies of functional food sample activity on production parameters. In this case, the most valuable turned out to be 36% mc and 60 rpm with 10% and 30% fresh kale contents. 

It is worth mentioning that there are significant differences between the activity of the extracts evaluated with DPPH and FRAP methods. In the case of DPPH (TEAC value), the highest activity was observed for the following samples: 10% of the additive, 36 mc, and 60 and 100 rpm, as well as 30% of the additive, 36 mc, and 100 rpm. FRAP values were the most efficient for all samples with the 30% kale additive. 

In the next phase of the experiment, the total polyphenolic compound content (TPC) was examined by applying the Folin–Ciocalteu method. The outcome of this work demonstrated that the addition of kale brought about strong pharmacological properties. The results showed that the phenolic compound content increased significantly due to the increase of the fresh kale addition (Table 3) for both 60 and 100 rpm screw speeds, and it was generally higher for samples processed at 100 rpm due to a shorter residence time inside the extruder. The addition of fresh kale engendered a significant increase in polyphenol content in comparison to control (potato) samples. Of note, variable production parameters had significant influence on the TPC. The best TPC values were obtained for different production parameters for each fresh kale content, such as 32% mc and 100 rpm for the 10% content of fresh kale and 36% mc and 100 rpm for the 30% fresh kale addition.

The obtained results of antioxidant activity (DPPH), reduction activity (FRAP), and total phenolic content (TPC) were correlated (Table 4). 

We then determined phenolic acid content. This was performed using the sample revealing the highest phenolic content, most advantageous chemical composition, and high antioxidant properties. The production parameters were the following: fresh kale content of 30%, 36% mc, and 100 rpm. The following phenolic acids were identified in this sample: protocatechuic, 4-OH-benzoic, vanillic, syringic, salicylic (benzoic acid derivatives), and caffeic, coumaric, ferulic, sinapic (cinnamic acid derivatives) (Table 5). Sinapic acid was the prevailing phenolic acid. 

Fresh kale addition caused a significant increase in antioxidant activity and total phenolic content. The level of activity was strongly connected to the content of the plant additive; nevertheless, even 10% of the additive led to an increase in the free radical scavenging ability. The level of antioxidant activity is important due to the fact that oxidative stress, being a consequence of imbalance of free radicals and antioxidants in the organism, leads to various harmful changes in cells and the development of various disorders [13,14]. Plants are inexhaustible sources of natural antioxidants as secondary plant metabolites, including the aforementioned polyphenolic compounds. The higher the content of these compounds, the higher the observed pro-health value. Considering the obtained study results, it is evident that an increase in plant additive content brought about an increase in antioxidant activity within the functional food samples. Similar study results were obtained for extruded corn snacks enriched with fresh kale [12]. Our results also explicitly indicated the high antioxidant activity of the functional food samples even with a minor addition of fresh kale (4, 6, and 8%). In comparison to the presented outcome based on potato blends, the antioxidant activity was higher in the studies presented by Kasprzak et al. [12]. The differences probably resulted from the high antioxidant activity of corn (the basis for food samples in this study), which equaled 40% after 30 min of measurement. In contrast, the activity of potato snack pellets was equal to 22% after the same amount of time.

It is worth mentioning the differences in activity obtained for analyzed samples with the use of DPPH and FRAP methods. The differences can result from the various mechanism reactions of the methods. It is known that the DPPH method is based on the reaction with free radical scavengers that are able to donate an electron or hydrogen to the stable DPPH free radical. In the case of the FRAP method, the reaction is based on the reduction of Fe^3+^ to Fe^2+^ through the donation of an electron. Hence, the mechanisms of action of the two reactions are slightly different in what can impact differences in the obtained activity. The two parameters should not be compared directly and should be used to indicate the most active samples for each parameter separately. The antioxidant capacity of a biological sample cannot be evaluated by a dingle assay since many factors are not taken into consideration [15,16].

In accordance with the HPLC-MS analysis [14], the most abundant individual phenolic acids in fresh kale leaf extracts were caffeic and ferulic acids (4887 ng/g d.w. and 4269 ng/g d.w., respectively). A similar correlation was observed for the extruded snack pellets supplemented with a fresh kale addition. The highest content of phenolic acids was recorded for caffeic acid, followed by sinapic and ferulic acids. Over all, the value of phenolic acid content within the processed samples was raised due to the higher content of these compounds in the fresh kale addition.

Production parameters also had a significant influence upon the sample values. The comparatively proportional increase in the antioxidant activity and TPC with the addition of fresh kale demonstrates that the extrusion-cooking process (under conditions of high temperature and high pressure) did not degrade the active phenolic compounds present in snack pellets. In this regard, slight deviations may indicate that the ingredients had not been mixed properly during the preparation at the laboratory scale. Literature research indicates that the polyphenol level and antioxidant activity of the product may vary due to the food manufacturing parameters [17]. In the case of free radical scavenging activity, the most favorable turned out to be the highest moisture (36%) and screw rotation speed (100 rpm), for which the highest activity was observed in samples with a 30% fresh kale content. Indeed, Multari et al. [18] showed that high-temperature treatment can enhance the release of phenolic compounds bound to cell wall structures.

The intense processing during extrusion-cooking results in the deep transformation of individual components [19,20]. Khanal et al. [21] showed the effects of extrusion-cooking on polyphenol content in grape seed and pomace. These authors demonstrated that this high-temperature-short-time method increases the levels of low-molecular-weight compounds (e.g., procyanidin) and releases biologically active monomers and dimers from polymer chains [21]. The intensity of changes therein depends on the properties of the raw material and the processing parameters. Properly chosen extrusion-cooking conditions (e.g., temperature, screw speed, moisture content, homogenization, and residence time distribution) may therefore release phenolic acids from the chemical bonds that they create with other compounds without deactivating aglycones [19,20,21]. This is due to the cracking of the rigid plant tissue components [22].

### 2.2. Physical Properties of Snack Pellets Enriched with Fresh Kale

Selected physical properties of snack pellets enriched with fresh kale are presented in Table 6.

Extrusion-cooking is a thermos-mechanical processing operation that applies high temperatures on the processed material for a short time. In this treatment, many chemical reactions and functional changes occur in the extruded material. These lead to changes in its physical properties, nutritional value, and sensory characteristics and to reduction of the microbiological contamination [23]. Conclusions upon the quality of the derived products can be drawn by measuring several functional properties of the final product, e.g., water absorption index (WAI) and water solubility index (WSI). Table 4 shows the results of WAI measurements depending on the amount of additive and processing variables (the level of moistening of raw materials and the rotational screw speed of the extruder). The processing variables were observed to have an impact on the measurement results. After the extrusion of samples both without and with 10% of the additive, in samples with a 32% moistening level, a higher rotational speed of the extruder screw raised the WAI of the samples. The opposite trend was reported for samples with a 36% moisture level. The highest WAI measurement (3.92 g/g) was recorded in control samples without the addition of fresh kale, and it was processed at a 36% moisture level and 60 rpm. The lowest measurement result (2.93 g/g) was reported in a sample with 10% of the additive extruded at the highest process variables (36% mc and 100 rpm). For samples with 30% of the additive, higher rotational speeds reduced the WAI only in samples with a 36% moisture level. The samples extruded with 30% of fresh kale at the level of 36% initial moisture were characterized by the highest WAI index across the entire range of processing speeds.

When analyzing the results of control samples extruded without the additive, the WSI increased along with the increase of screw rotational speed at 32% mc. In most cases, an increase in the moisture level resulted in a higher WSI. The lowest WAI result (9.13%) was recorded in a sample extruded at the lowest process variables. The highest result (24.50%) was found in a sample with 10% of the additive extruded using the highest processing variables. For samples with 10% and 30% of the additive, we observed that a higher screw speed brought about higher WSI results. In samples extruded with 30% of the additive across the entire range of extruder screw speeds, samples with a 32% level of moistening showed the lowest WSI results.

The hydration capacity of raw materials and finished products is an important criteria of the quality. In the case of extrudates, WAI and WSI are characteristics that describe absorption and solubility when preparing these products with liquids. WSI and WAI values extend in a wide range depending on the conditions used during the extrusion-cooking process. Factors affecting the values of WAI and WSI can be classified into two types: the parameters related to the processed raw materials (raw material composition and formulation, the initial particle size of milled materials, the grinding procedure, and the pre-processing treatments) and factors linked to the extruder (equipment type, barrel and die temperatures, feed rate, residence time, screw configuration, speed and compression ratio, die dimension, and configuration) [24].

The WSI is related to the quantity of soluble molecules and is often used as an indicator of starch degradation. It depends on starch granule disruption, amylose, and amylopectin depolymerization, as well as starch gelatinization. These changes occur in the extruder under the influence of shear forces, pressure, and heat [25,26]. In general, the greater the percentage of amylose contained in the starch granules, the higher its solubility index. As already stated, many factors affect WAI and WSI values. WSI values of extrudates often decrease with the increasing content of dry functional ingredients (e.g., plants rich in polyphenols). This could be attributed to the degradation of starch and the breakdown of soluble fiber at a high energy expenditure of extrusion due to the low moisture content.

The findings of the study conducted by Medina-Rendon et al. [24] confirmed that, in most cases, WSI decreased with increased mango peel flour and mango kernel flour content during the production of extruded food based on white corn flour. Herein, the WSI was raised due to the disintegration of starch granules and the melting of low molecular compounds during the extrusion-cooking process. This caused an increase in soluble materials. Jin et al. [27] observed an increase in WSI of corn meal extrudates as the fiber level rose from 0 to 20%. This is due to the fact that water solubility gives information about degradation, and different results have been observed for the effect of the incorporation of food by-products on the extrudate functionality [27]. In the research reported by Oniszczuk et al. [28], WSI results showed the decreased solubility of extruded corn gruels with linden enrichment (from 15.32% for corn extrudates to 7.86% for the 10% additive level). Increasing the additive level enhanced the fiber content and decreased the starch content, reducing the solubility index.

It can be generally stated that a higher fiber content in enriched snacks results in the limiting of WAI due to the replacement of starch by fibrous fractions of plants. In our work, increased fiber content was responsible for lowering the WSI because of the low solubility of fibrous plant fractions during the test. Acosta-Pérez et al. [29] studied the effect of acetylation on the properties of rice starch and observed that acetylated rice starch had higher WAI and WSI values compared to native starch due to the introduction of acetyl groups in the modified rice starch. In addition, the presence of soluble fiber (containing mucilage, gums, pectins, and hemicellulose) can generate a more significant absorption and water solubility than starch content alone.

WAI indicates the amount of water immobilized by the material. In most cases, WAI decreases with the increased functional component content. This is attributed to decreasing starch content and, thus, less water absorption. In research conducted by Oniszczuk et al. [28], the WAI ranged from 5.51 g/g for corn gruels to 4.65 g/g for extrudates containing 10% linden flowers. The substitution for starchy materials of fruits, vegetables, or high-fiber additives causes the reduction of starch undergoing swelling and gelatinization during processing. Therefore, the WAI is usually much lower when additives are used. Of relevance, the initial water absorption characteristics of functional ingredients with high fiber contents and expansion ratios also affects the WAI. For example, the study results of Medina-Rendon et al. [26] revealed that the WAI increased as the percentage of mango peel flour in white corn flour increased during production of extruded functional food.

The moisture content of the used raw materials also has a significant impact on WAI. A high value of this parameter leads to gelatinization reactions, which in turn affect the extrudate physical properties and nutritional component stability. In contrast, a low moisture content can result in a dextrinization increase during HTST processing because of the relatively high viscosity and shear stress [30]. Liu et al. [30] found a WAI increase and a WSI decrease in extrudates when the extruded feed moisture content increased. This was not consistent with the findings of Ding et al. [31], wherein a lower WAI and higher WSI were evident upon an increase in the feed moisture content. The results in this study could be attributed to the fact that the high feed moisture content (20–25%) reduced the shear forces, thereby promoting starch dextrinization. 

The different results obtained by authors may be caused by the diverse materials used in experiments. Moreover, WAI is a temperature-dependent parameter. Altan et al. [32] reported that the WAI decreased from 7.03 to 6.10 g/g with the increasing temperature and pomace level during the examination of barley-tomato pomace extrudates. Campus-Baypoli et al. [33] observed an increase of WAI in the transformation of corn dough to tortilla and attributed higher WAI values in tortillas to the loss of starch granule structure and integrity, which leads to starch gelatinization. Likewise, high values in WAI observed in commercial flours can also be explained by the presence of hydrocolloids, where hydroxyl groups in their structure allow water interactions through hydrogen bonding.

In conclusion, the WAI and WSI are important parameters related to the quality of functional food. These values may depend on the expansion ratio and initial water absorption characteristics of raw materials, especially when ingredients with high fiber contents are added. However, these two properties are significantly influenced by the combined effects of raw materials and the processing conditions [34].

The results of fat absorption tests carried out after frying snacks from potato pellets with and without the addition of fresh kale processed under different extrusion conditions showed a varied effect of the applied parameters on fat absorption by the snacks (Table 4). Our research demonstrated that the use of different raw material blends and various rotational speeds of the extruder screw under the extrusion-cooking process had a significant impact on the value of the fat absorption index of fried snacks. In control samples without the addition of fresh kale, the fat absorption index increased along with the falling rotational speed of the extruder screw and with the rising initial moisture content of the raw material blends undergoing the extrusion-cooking process. These samples returned the highest values of fat absorption in all tests, the greatest value being 47.62% for fried snacks obtained from pellets processed at a moisture content of 32% and extruded at 60 rpm.

In fried snacks obtained from pellets with the addition of fresh kale (10 and 30%), fat absorption was enhanced for snacks made of half-product obtained at low speed of the extruder screw. For snacks fried from pellets with the addition of fresh kale, the fat absorption index was lower for the majority of products obtained from blends with a high level of moisture content before the extrusion-cooking process. The fat absorption value for fried snacks ranged from 5 to 47%. In general, lower fat absorption was reported along with the higher addition of fresh kale, and at the same time, a greater water content, which disturbed the expansion process, thus reduced fat absorption by the pellets during frying expansion. The reported dependencies are similar to the results achieved by Matysiak et al. [35].

The fat absorption index is connected to the flavor and mouthfeel by entrapment of oil [36]. Water binds the hydrophilic groups via hydrogen bonds, and oil binds the hydrophobic groups of the protein chains. Therefore, different functional additives can have different effects on the final fat absorption index of extruded food products. In the production of feed based on incorporating pea protein isolate with a 0 to 40% addition of mycelium, fat absorption increased significantly with the addition of mycelium in both raw as well as extruded samples. This was due to the fibrous and porous structure of mycelium, which holds the oil in its pores [37]. The fat absorption index content in this feed ranged from 130% to 178%. The results of the fat absorption were in agreement with Samard and Ryu [38] and Lam et al. [39] for isolated pea protein.

### 2.3. The Basic Composition and Fatty Acids Profile

In our study, we also determined the basic composition and fatty acid profile of extrudates with the addition of fresh kale. The two varied significantly in the produced snack pellets. The differences were attributed to the varied content of fresh kale used as a component for the production of the extrudates. 

With the addition of 30% of fresh kale to the extrudates, an increase in the content of proximate components was observed in relation to those containing 10% fresh kale, as well as control samples without it, for both 60 and 100 rpm screw speeds. The content of the following basic nutritional components was reported to rise: ash (and increase in the amount of micro- and macroelements as the result of this), total protein, fat, and fiber, the latter being a desirable nutritional component of snacks (Table 7). An evident increase in the content of total protein in snacks with the highest addition of fresh kale undoubtedly enhances their nutritional value. 

In snack pellets without a fresh kale addition, the component content was the lowest. Here, the protein content reached 3.46–3.64 g/100 g, ash content was 3.70–3.76 g/100 g, fat was 0.07–0.11 g/100 g, fiber was 5.18–5.78 g/100 g, and carbohydrate content was 76.09–77.37 g/100 g. Higher values were obtained for snack pellets with fresh kale additions. Here, protein content reached 4.08–6.41 g/100 g, ash content was 4.19–5.64 g/100 g, fat was 0.12–0.36 g/100 g, fiber was 5.92–7.22 g/100 g, and carbohydrate content was 71.02–75.55 g/100 g (Table 7).

The content of fiber in extruded snack pellets (lower than expected) may be affected by the degradation of macromolecular compounds as a result of the extrusion-cooking processing and the associated impact of high temperature and pressure on the material during treatment. Fresh kale is a low-calorie vegetable with a low fat content (0.93 g/100 g d.w.); thus, any food enriched with fresh kale is an ideal part of a reduction diet due to the low amount of fat after processing. It is also a fairly rich source of vegetable protein (4.28 g/100 g d.w.). The healthy nutritional traits, low fat content, and high content of protein highlight the potential of fresh kale as a functional food ingredient [40].

The increasing content of fresh kale from 10 to 30% as a component in the production of extruded snack pellets caused an increase in the share of PUFA in the total amount of fatty acids, which clearly enhanced the nutritional content of this type of snack pellets (Table 8 and Table 9). What is more, while the extrusion-cooking process may reduce the content of bioactive substances, at the same time, it may often increase their bioavailability in the human body, which translates into an extended actual volume of absorbed bioactive compounds. 

Research results obtained by Nemzer et al. [40] demonstrated that fresh kale is a rich source of fatty acids (23.61 mg/100 g dry weight, d.w.), especially polyunsaturated fatty acids, i.e., PUFA (19.21 mg/100 g d.w.). In their work, they compared fatty acids in spinach and fresh kale and revealed significant differences (<0.05) in the C16:0 content of 2.99 ± 0.25 mg/100 g d.w. and 4.40 ± 0.18 mg/100 g d.w., respectively. Additionally, when comparing omega-3 (C18:3n-3), spinach (10.84 ± 0.86 mg/100 g d.w.), and fresh kale (16.69 ± 2.44 mg/100 g d.w.), kale contained significantly different (*p* < 0.05) amounts. However, no significant differences were observed for omega-6 (C18:2n-6) fatty acids between spinach (3.48 ± 0.43 mg/100 g d.w.) and fresh kale (3.54 ± 0.61 mg/100 g d.w.).

### 2.4. Principal Component Analysis (PCA)

We undertook principal component analysis to show the relationships between selected properties of snack pellets with the addition of fresh kale. In the diagram (Figure 1 and Figure 2), the parameters indicated between the two red circles had the greatest impact on the variability of the system (i.e., all features, Figure 1). Fiber, protein, and ash content were positively correlated with one another. There is also a positive and strong correlation between WAI and fat absorption index. In contrast, a strong and negative correlation can be seen between fat content and WAI and fat absorption index. The control sample (0%) value had the greatest impact on WAI and fat absorption index, and the 10% addition of fresh kale to snack pellets had the greatest impact on fat content and WSI. For samples with the 30% addition of fresh kale, significant effects on fiber, protein, and ash content were determined (Figure 1 and Figure 2).

The PCA plots reveal that the first principal component (PC1) refers to the use of fresh kale addition at 55.36%. In the figure, positive values of PC1 describe the results obtained for control snack pellets without fresh kale. The green zone (circle) group parameters refer to the product without the additive (Figure 1 and Figure 2). The negative values of PC1 describe the extruded products with 30% of fresh kale and the parameters that had an impact on fiber, protein, and ash content. The properties of the snack pellet extrudates produced without the addition of fresh kale or with the addition of 30% fresh kale were similar, as they found similar space in the PC1 and PC2 loading plot (Figure 3), while a large dispersion of the analysis results of individual components was observed for extrudates with the addition of 10% fresh kale under various processing conditions for the production of snack pellets.

## 3. Materials and Methods

### 3.1. Samples Preparation

The research material was snack pellets made on the basis of various recipes. The pellets were a blend of the following potato components: potato starch (SUPERIOR STANDARD, Enterprise of Potato Industry Bronislaw S.A., Poland), potato flakes (supplier: Potato Industry in Lublin, Poland), and potato grits (supplier: Potato Industry in Lublin, Poland). In addition to the raw material blends, fresh kale from organic crops was used as an additive (supplier: ANREKO Andrzej Gębka, Niemce, Poland) in the amounts of 10% and 30% of the total blend. The fresh kale was first shredded with a blender. Each of the pellet blends was prepared with modified proportions of ingredients and water depending on the amount of plant additive used. Blends of potato components with fresh kale were kept for at least 0.5 h before extrusion to ensure uniform moisture distribution. Before the extrusion-cooking process, the moisture content of all blends was assessed using a standard drying procedure (130 °C for 1 h), and when needed, all blends were moistened to the intended value of 32% and 36% of moisture content (mc). 

The extrusion-cooking process was carried out in a prototype single-screw extruder EXP-45-32 (Zamak Mercator, Skawina, Poland), where the length of the working barrel to the screw diameter was L/D = 20. The tests were carried out at variable screw rotations (60 and 100 rpm). The obtained extrudates were formed by a roller system with a cooling system (fans) installed at the cutting system end, which used high-speed knives to shape the extrudates into 25 × 25 mm squares. Such shaped pellets were carefully transported to a laboratory dryer (laboratory shelf dryer (own design) with forced air circulation by two fans and an incorporated 2 kW heating system; drying temperature was maximum 40 °C) in order to reduce the moisture content of the finished product below 12%. The resulting snack pellets and, later, fried snacks (preparation described in 3.9) were kept in dry and dark conditions and were shredded with a laboratory grinder for particles smaller than 0.3 mm in diameter for further testing. A laboratory grinder, LMN100 (TestChem, Radlin, Poland), was used in experimental preparation.

### 3.2. Preparation of Extracts

The extracts were prepared with the use of an ultrasonic bath (Bandelin Electronic GmbH & Co., KG, Berlin, Germany) with the following parameters: a temperature of 60 °C, an ultrasound frequency of 33 kHz, and a power of 320 W [12]. In order to obtain the test extracts, 4 g of ground extrudes was mixed with 80 mL of aqueous methanol (80 mL methanol: 20 mL H_2_O *v*/*v*) and placed in the ultrasonic bath for 40 min. After this time, the extracts were filtered through a paper filter and a new portion (80 mL) of methanol was added to the remainder to repeat the extraction process. The obtained portions of extracts were combined, evaporated to dryness, and dissolved in 10 mL of methanol. The samples were examined regarding their free radical scavenging activity, total phenolic content, and level of free phenolic acids. The extracts were then filtered through a 0.45 μm nylon syringe filter (Millex-HN, Ireland) before chromatographic analysis.

### 3.3. Free Radical Scavenging Activity—DPPH Method

We applied the DPPH (2,2-diphenyl-1-picrylhydrazyl) method in order to determine the free radical scavenging activity of the obtained extracts. The studies were performed based on a modified method originated by Burda and Oleszek [41] with the use of a UV-VIS spectrophotometer Genesys 20 UV-VIST (Thermo Scientific, Waltham, MA, USA). The following parameters were used: 517 nm wavelength, measurements every 5 min for 30 min, and calibration based on pure methanol. All measurements were repeated three times. The obtained results are presented as RSA and TEAC values.

### 3.4. Ferric-Reducing Antioxidant Power (FRAP)

The FRAP reagent was made from solutions prepared in the following way. Acetate buffer (0.3 M, pH 3.6) was mixed with aqueous iron (III) chloride solution (0.02 M) and TPTZ solution (0.01 M) in a 10:1:1 ratio, respectively; then, 500 μL of each extract was taken into separate vials, and 2.5 mL of FRAP basic reagent was added. The vessels were closed, mixed, and placed in a 37 °C water bath for 30 min. Absorbance was measured using a UV-VIS spectrophotometer at 593 nm against a control, in which the extract was replaced with methanol. Each measurement was performed in triplicate. The obtained results are presented as FRAP units (Fe^2+^ μg/mL) and gallic acid equivalents (μg/mL).

### 3.5. Total Content of Polyphenolic Compounds (TPC) with Use of Folin-Ciocalteu Method

The measurement was performed using a modified method employing the Folin–Ciocalteu (FC) reagent that was put forward in [12]. The following procedure was applied: 200 μL of extract was mixed with 1.8 mL of water. The Folin reagent (200 μL) was then added, and the mixture was vigorously mixed. Five min after the reaction initiation, 2 mL of 7% Na_2_CO_3_ was added. The mixture was subsequently incubated at 40 °C for 60 min. Absorbance was measured with the use of a UV-VIS spectrophotometer at 760 nm. The amount of total phenolics was expressed as μg of gallic acid equivalents (GAE) per g of dry mass (d.m.).

### 3.6. Content of Phenolic Acids

The phenolic acid content was analyzed utilizing a Waters ACQUITY UPLC Chromatograph (Waters Corp., Milford, MA, USA), equipped with a PDA and a triple-quadrupole mass detector (Waters Corp., Milford, MA, USA). Samples (50 mg/mL) were first separated on a Waters ACQUITY UPLC^®^ HSS C18 column (100 × 2.1 mm, 1.8 μm) at 30 °C. The mobile phase consisted of solvent A (0.1% formic acid in water) and solvent B (acetonitrile with 0.1% formic acid). The elution step (0.50 mL/min) was carried out with a gradient of solvent B: 0–0.5 min, 8% B; 0.5–8 min, 8–20% B; 8–8.10 min, 20–95% B; 8.10–10 min, 95% B; 10–10.10%, 95–8% B; 10.10–12 min, 8% B. The sample injection volume was 2.5 μL (full loop mode). The detection of phenolic acids was performed in the negative ionization mode by applying a selected reaction monitoring method. The condition of the MS analysis was published by Czaban et al. [42]. The concentration of phenolic acids in snack pellet extracts was calculated on the basis of calibration curves (Table 10).

### 3.7. Water Absorption Index (WAI) 

WAI was determined as described previously [43]. A suspension was prepared from ground pellets of 0.7 g and 7 mL of distilled water by continuous mixing for 20 min. The suspension was then centrifuged at 15,000 rpm for 10 min in a Digicen 21 laboratory centrifuge (Labsystem, Kraków, Poland). Filtrate was subsequently collected from over the obtained gel, and next, the gel was weighed. The water absorption index WAI was calculated using the following formula:(1)$$\mathrm{WAI}=\frac{m_{z}}{m_{s}}100\%\;[g/g]\;$$
where m*_s_* is the dry sample mass (g); m*_z_* is the gel mass (g).

### 3.8. Water Solubility Index (WSI)

WSI was determined as described previously [43]. The filtrate obtained after the measurement of the WAI was dried at 130 °C until the water evaporated completely. The water solubility index was calculated using the following formula:(2)$$\mathrm{WSI}=\frac{m_{s}-m_{ps}}{m_{pp}}\;100\;[\%]$$
where m*_s_* is the vessel mass after drying (g), m*_p__s_* is the vessel mass before drying (g), and m*_pp_* is the sample mass (g).

### 3.9. Fat Absorption Index

The fat Absorption Index was determined as described previously [43]. To determine the absorption index of fat, samples in the form of dried snack pellets with an initial weight of P*_p_* were fried in vegetable oil at 190 °C until the product had expanded properly and came up to the surface. The fried product was drained and placed on a laboratory scale to measure the final weight (P*_o_*). The fat absorption index was calculated using the following formula:(3)$$P=\frac{P_{o}-P_{p}}{P_{p}}\times 100\left[\%\right]$$
where P*_p_* is the pellet mass before frying (g); P is the snack mass after frying and draining the oil (g). All physical properties were performed in three replications.

### 3.10. Chemical Composition

The content of dry matter and basic nutrients in ground samples (250 g of each accession) was determined according to standard AACC [44] and AOAC [45] procedures. Protein (AACC 46–10), fat (AACC 30–10), and ash (AACC 08–01) contents were determined in triplicate [44]. The content of fiber was tested using the 993.21 method [45]. The content of available carbohydrates was calculated as dry matter–protein–fat–ash–fiber. 

The fatty acid composition was assessed by applying the gas chromatography method on a Varian CP-3800 chromatograph CP-3800 (Varian Inc., Palo Alto, CA, USA) after the conversion of fats to fatty acid methyl esters (FAME) according to the AOAC method [44]. The chromatograph operating conditions for fatty acid separation were as follows: capillary column CP WAX 52CB DF 0.25 mm of 60 m length; gas carrier, helium; flow rate, 1.4 mL min^−1^; column temperature, +120 °C gradually increasing by 20 °C min^−1^; determination time, 127 min; feeder temperature, 160 °C; detector temperature, 160 °C; other gases, hydrogen and oxygen. The determinations were based on a Supelco 37 Component Fame Mix template (Sigma-Aldrich, Poznań, Poland). The results were expressed as the proportion of individual fatty acids in the total value of fatty acids taken as 100%.

### 3.11. Statistical Analysis

Statistica software (v. 12.0, StatSoft Inc., Tulsa, OK, USA) was used for all statistical analyses. Principal component analysis (PCA) was applied to define the relationship between the addition of fresh kale (%), the screw rotation (rpm), and the moisture (mc) and physicochemical properties of snack pellets. The input matrix was scaled up automatically. The optimal number of principal components obtained in the analysis was established using Cattell’s criterion.

## 4. Conclusions

This paper describes studies of newly developed innovative snack pellets incorporating 10 and 30% of fresh kale, a plant rich in polyphenols. The addition of this raw material caused significant increases in antioxidant activity and in total phenolic content. The production parameters also showed significant influence on the antioxidant activity. Screw speed and moisture content were found to be crucial for the polyphenol content and antioxidant activity of the product. Herein, properly chosen extrusion-cooking conditions may release phenolic acids from the chemical bonds that they create with other compounds without deactivating aglycones. Processing conditions also displayed significant influence on the water absorption and water solubility index values, wherein WAI and WSI are important parameters related to the quality of the extruded functional food. Our research indicated that the use of different blend compositions in the extrusion process, as well as various extruder screw speeds and diverse moisture contents of the raw materials, had significant impacts on physical properties, especially on the fat absorption index. Our experiment demonstrated a lowering from 47% to 5% if the level of fresh kale increased and (at the same time) water content increased. This disturbed the expansion process, thus reducing fat absorption by the pellets during frying. Along with the addition of 30% of fresh kale to the extrudates, an increase in the content of nutritional components was observed in relation to those containing 10% of fresh kale and to the control sample produced without it, for both 60 and 100 rpm screw speeds. The healthy nutritional traits, antioxidant activity, low fat content, and high content of protein highlight the potential of snack pellets enriched with fresh kale as a functional food product.

## Figures and Tables

**Figure 1 molecules-28-01835-f001:**
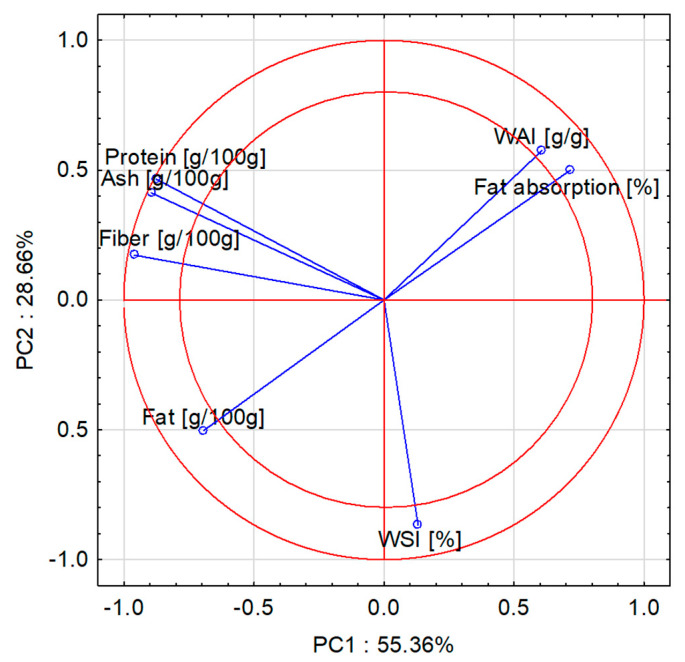
PC1 and PC2 score plot. Projection of variables: fiber, protein, ash, fat, WAI, WSI, and fat absorption index.

**Figure 2 molecules-28-01835-f002:**
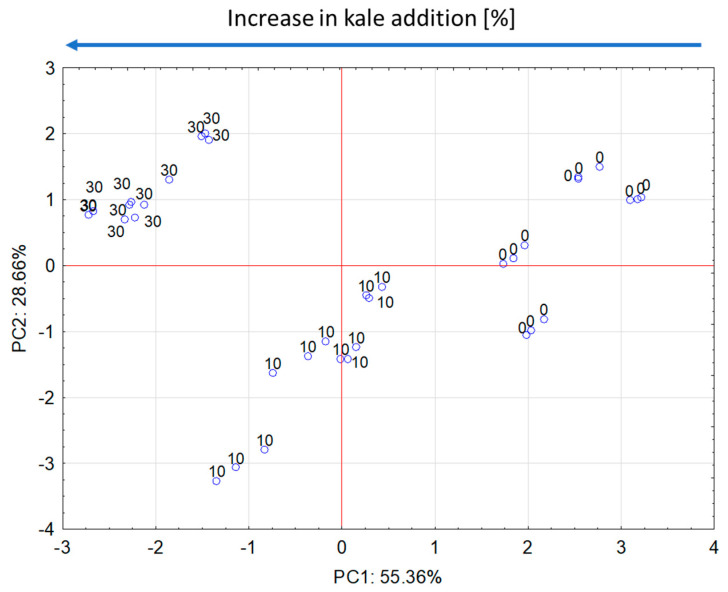
PC1 and PC2 loading plot. Projection of cases characterizing the level of fresh kale addition.

**Figure 3 molecules-28-01835-f003:**
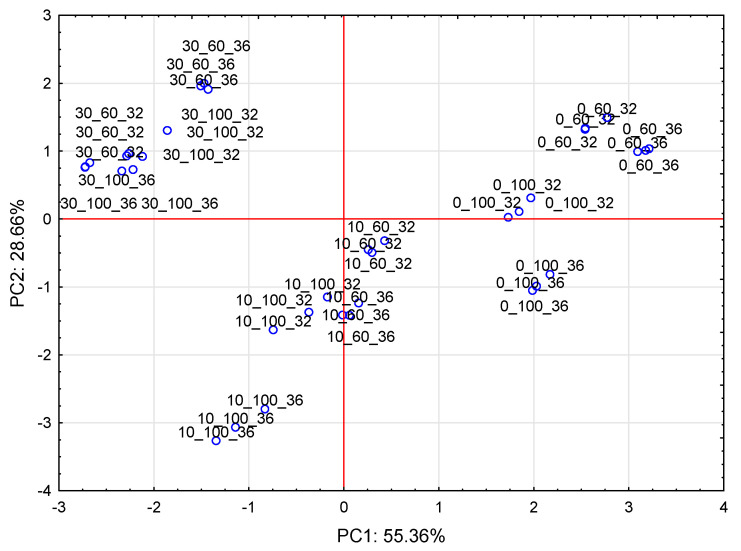
PC1 and PC2 loading plot. Projection of cases characterizing the level of fresh kale addition with screw rotation and moisture.

**Table 1 molecules-28-01835-t001:** TEAC value (μg/g product) calculated for all analyzed extract samples at 10 min from the DPPH reaction initiation.

Extract’s Samples	TEAC Value [μg/g Product]
0%, 32 mc, 60 rpm	73.83 ± 3.04
0%, 32 mc, 100 rpm	73.25 ± 2.09
0%, 36 mc, 60 rpm	73.36 ± 4,02
0%, 36 mc, 100 rpm	73.6 ± 2.87
10%, 32 mc, 60 rpm	78.39 ± 2.07
10%, 32 mc, 100 rpm	95.44 ± 4.67
10%, 36 mc, 60 rpm	108.88 ± 4.99
10%, 36 mc, 100 rpm	105.96 ± 4.87
30%, 32 mc, 60 rpm	98.71 ± 3.45
30%, 32 mc, 100 rpm	99.07 ± 3.00
30%, 36 mc, 60 rpm	97.31 ± 5.03
30%, 36 mc, 100 rpm	109.81 ± 5.00

**Table 2 molecules-28-01835-t002:** Parameter values describing extract antioxidant properties (n = 3, ± SD).

Fresh Kale Content (%)	Screw Speed (rpm)	Moisture Content (%)	Activity towards Fe^3+^ Reduction	
			FRAP Units (Fe^2+^ μg/mL)	Gallic Acid Equivalent (μg/mL)
0	60	32	4.56 ± 0.06	1.46 ± 0.03
		36	2.39 ± 0.02	0.76 ± 0.01
	100	32	3.73 ± 0.04	1.19 ± 0.01
		36	3.22 ± 0.02	1.03 ± 0.02
10	60	32	4.72 ± 0.02	1.51 ± 0.01
		36	5.44 ± 0.02	1.74 ± 0.02
	100	32	4.34 ± 0.01	1.39 ± 0.02
		36	3.77 ± 0.02	1.21 ± 0.01
30	60	32	6.43 ± 0.02	2.21 ± 0.00
		36	7.52 ± 0.08	2.42 ± 0.03
	100	32	6.27 ± 0.05	2.01 ± 0.01
		36	7.44 ± 0.09	2.39 ± 0.02

**Table 3 molecules-28-01835-t003:** Total content of polyphenolic compounds (TPC) in snack pellets enriched with fresh kale (n = 3, ± SD).

Fresh Kale Content (%)	Screw Speed (rpm)	Moisture Content (%)	Total Phenolic Content (μg GAE/g d.w.)
0	60	32	19.1 ± 0.12
		36	21.7 ± 0.07
	100	32	23.1 ± 0.06
		36	21.8 ± 0.05
10	60	32	27.1 ± 0.05
		36	32.5 ± 0.04
	100	32	35.2 ± 0.09
		36	25.8 ± 0.10
30	60	32	62.1 ± 0.17
		36	66.1 ± 0.17
	100	32	70.1 ± 0.09
		36	72.8 ± 0.08

**Table 4 molecules-28-01835-t004:** Correlation coefficients for DPPH, FRAP, and TPC.

	Correlation Coefficients		
Sample Parameters	DPPH	FRAP	TPC
0%, 10%, 36% 60 rpm, 32 mc	0.987	0.967	0.988
0%, 10%, 36% 60 rpm, 36 mc	0.782	0.997	0.996
0%, 10%, 36% 100 rpm, 32 mc	0.834	0.955	0.955
0%, 10%, 36% 100 rpm, 36 mc	0.816	0.977	0.965

**Table 5 molecules-28-01835-t005:** Phenolic acid contents in snack pellets with a 30% content of fresh kale (36% mc, 100 rpm) (n = 3; mean ± SD).

Phenolic Acid	Content of Phenolic Acid (μg/g)
protocatechuic	0.304 ± 0.088
p-OH-benzoic	0.922 ± 0.168
vanillic	BLOQ
caffeic	30.053 ± 2.819
syringic	BLOQ
p-coumaric	3.208 ± 0.342
ferulic	21.376 ± 2.292
sinapic	30.144 ± 1.679
salicylic	0.338 ± 0.014

BLOQ, result below Limit of Quantification and above the Limit of Detection.

**Table 6 molecules-28-01835-t006:** Selected physical properties of snack pellets enriched with fresh kale (n = 3, ± SD).

Fresh Kale Content (%)	Screw Speed (rpm)	Moisture Content (%)	WAI (g/g)	WSI (%)	Fat Absorption Index (%)
0	60	32	3.57 ± 0.26	9.13 ± 0.42	34.93 ± 7.12
		36	3.92 ± 0.18	15.43 ± 0.62	36.08 ± 4.02
	100	32	3.70 ± 0.16	14.87 ± 0.56	18.45 ± 6.23
		36	3.50 ± 0.22	23.40 ± 0.68	22.69 ± 2.64
10	60	32	2.97 ± 0.14	14.84 ± 0.34	14.67 ± 2.58
		36	2.99 ± 0.16	19.86 ± 0.58	16.76 ± 1.45
	100	32	3.06 ± 0.24	18.39 ± 0.48	5.75 ± 2.64
		36	2.93 ± 0.32	24.50 ± 0.52	6.13 ± 2.12
30	60	32	2.99 ± 0.12	10.84 ± 0.23	8.68 ± 2.42
		36	3.50 ± 0.07	13.12 ± 0.34	20.90 ± 1.81
	100	32	3.28 ± 0.15	11.27 ± 0.22	12.22 ± 5.55
		36	3.36 ± 0.19	13.96 ± 0.28	12.42 ± 2.25

WAI, water absorption index; WSI, water solubility index.

**Table 7 molecules-28-01835-t007:** Basic composition of snack pellets with the addition of fresh kale (n = 3, ± SD).

Fresh Kale Content (%)	Screw Speed (rpm)	Moisture Content (%)	Component (g/100 g)					
			Dry Matter	Protein	Ash	Fat	Fiber	Carbohydrates
0	60	32	89.94 ± 2.09	3.64 ± 0.87	3.76 ± 0.62	0.07 ± 0.01	5.24 ± 1.08	77.23 ± 2.45
		36	89.98 ± 2.52	3.56 ± 0.86	3.76 ± 0.58	0.11 ± 0.01	5.18 ± 0.98	77.37 ± 2.69
	100	32	89.21 ± 2.66	3.57 ± 0.89	3.76 ± 0.46	0.09 ± 0.01	5.64 ± 0.88	76.15 ± 2.98
		36	89.14 ± 2.87	3.46 ± 0.78	3.70 ± 0.38	0.11 ± 0.01	5.78 ± 0.78	76.09 ± 2.89
10	60	32	90.09 ± 2.32	4.31 ± 0.78	4.19 ± 0.48	0.12 ± 0.01	5.92 ± 0.89	75.55 ± 2.74
		36	89.81 ± 1.98	4.11 ± 0.88	4.20 ± 0.78	0.21 ± 0.01	6.12 ± 1.02	75.17 ± 3.02
	100	32	90.22 ± 2.98	4.32 ± 0.78	4.19 ± 0.34	0.19 ± 0.01	6.22 ± 0.88	75.30 ± 3.06
		36	89.70 ± 2.74	4.08 ± 0.74	4.26 ± 0.52	0.36 ± 0.02	6.26 ± 1.04	74.77 ± 3.12
30	60	32	90.25 ± 2.71	6.06 ± 0.83	5.32 ± 0.68	0.19 ± 0.01	6.92 ± 1.06	71.76 ± 3.45
		36	90.73 ± 2.66	6.39 ± 0.79	5.57 ± 0.59	0.15 ± 0.01	7.06 ± 0.98	71.56 ± 3.65
	100	32	90.19 ± 2.45	6.19 ± 0.82	5.22 ± 0.49	0.21 ± 0.01	7.12 ± 0.88	71.45 ± 2.96
		36	90.57 ± 2.32	6.41 ± 0.83	5.64 ± 0.47	0.28 ± 0.02	7.22 ± 0.96	71.02 ± 2.94

**Table 8 molecules-28-01835-t008:** Fatty acid profiles of snack pellets processed at a screw speed of 60 rpm.

Fatty Acid	Content of Fatty Acids (mg/100 g)			
	10%; 32 mc; 60 rpm	10%; 36 mc; 60 rpm	30%; 32 mc; 60 rpm	30%; 36 mc; 60 rpm
Myrystic acid, C 14:0	0.000 ± 0.000	0.000 ± 0.000	0.000 ± 0.000	2.992 ± 0.232
Pentadecylic acid, C 15:0	0.000 ± 0.000	0.000 ± 0.000	0.000 ± 0.000	0.000 ± 0.000
Palmitic acid, C 16:0	8.212 ± 0.568	17.754 ± 1.232	8.147 ± 0.623	9.275 ± 0.689
Palmitoleic acid, C 16:1n-7	0.000 ± 0.000	6.604 ± 0.524	4.958 ± 0.428	0.000 ± 0.000
Stearic acid, C 18:0	4.504 ± 0.523	8.267 ± 0.952	4.478 ± 0.597	3.430 ± 0.128
Oleic acid, C 18:1n-9	57.177 ± 4.658	46.975 ± 3.865	46.422 ± 3.895	41.003 ± 3.126
Linoleic acid, C 18:2n-6	19.456 ± 1.784	15.626 ± 1.256	21.418 ± 2.056	21.977 ± 2.254
*α*-*Linolenic acid*, C 18:3n-3	10.651 ± 0.856	4.773 ± 0.423	14.578 ± 1.298	21.324 ± 2.078
Arachidic acid, C 20:0	0.000 ± 0.000	0.000 ± 0.000	0.000 ± 0.000	0.000 ± 0.000
Gondoic acid, C 20:1n-9	0.000 ± 0.000	0.000 ± 0.000	0.000 ± 0.000	0.000 ± 0.000
TOTAL	100	99.999	100.001	100.001
ΣSFA	12.716	26.021	12.625	15.697
ΣMUFA	57.177	53.579	51.380	41.003
ΣPUFA	30.107	20.399	35.996	43.301

SFA, saturated fatty acids; MUFA, *monounsaturated* fatty acid; PUFA, polyunsaturated fatty acids.

**Table 9 molecules-28-01835-t009:** Fatty acid profiles of snack pellets processed at a screw speed of 100 rpm.

Fatty Acid	Content of Fatty Acids (mg/100 g)			
	10%; 32 mc; 100 rpm	10%; 36 mc; 100 rpm	30%; 32 mc; 100 rpm	30%; 36 mc; 100 rpm
Myrystic acid, C 14:0	0.000 ± 0.000	0.000 ± 0.000	1.015 ± 0.189	1.451 ± 0.125
Pentadecylic acid, C 15:0	0.000 ± 0.000	0.000 ± 0.000	0.360 ± 0.042	0.443 ± 0.052
Palmitic acid, C 16:0	11.343 ± 0.985	10.872 ± 0.852	10.247 ± 0.789	11.716 ± 1.005
Palmitoleic acid, C 16:1n-7	0.000 ± 0.000	3.746 ± 0.254	0.000 ± 0.000	0.000 ± 0.000
Stearic acid, C 18:0	4.434 ± 0.545	6.563 ± 0.625	3.458 ± 0.278	4.040 ± 0.325
Oleic acid, C 18:1n-9	52.614 ± 4.874	53.279 ± 5.025	44.974 ± 3.897	43.431 ± 3.874
Linoleic acid, C 18:2n-6	20.197 ± 1.854	19.619 ± 1.254	20.998 ± 1.523	20.720 ± 1.874
*α*-*Linolenic acid,* C 18:3n-3	11.412 ± 0.854	5.922 ± 0.252	17.290 ± 1.546	16.332 ± 1.532
Arachidic acid, C 20:0	0.000 ± 0.000	0.000 ± 0.000	0.890 ± 0.052	0.615 ± 0.054
Gondoic acid, C 20:1n-9	0.000 ± 0.000	0.000 ± 0.000	0.768 ± 0.050	1.252 ± 0.098
TOTAL	100	100.001	100	100
ΣSFA	15.777	17.435	15.97	18.265
ΣMUFA	52.614	57.025	45.742	44.683
ΣPUFA	31.609	25.541	38.288	37.052

SFA, saturated fatty acids; MUFA. *monounsaturated* fatty acid; PUFA, polyunsaturated fatty acids.

**Table 10 molecules-28-01835-t010:** Parameters of the calibration curve for nine different phenolic acids.

Phenolic Acid	Calibration Curve	R^2^
protocatechuic	y = −0.02544x^2^ + 1.46612x + 0.01376	0.997
p-OH-benzoic	y = −0.01168x^2^ + 1.43904x + 0.16496	0.997
vanillic	y = 0.00011x^2^ +0.194029x − 0.00311	0.998
caffeic	y = −0.01827x^2^ + 2.42109x + 0.43679	0.995
syringic	y = −0.00005x^2^ +0.259824x − 0.00264	0.986
p-coumaric	y = −0.01657x^2^ + 2.05818x + 2.05818	0.993
ferulic	y = −0.00041x^2^ + 0.380126x + 3.30005	0.994
sinapic	y = −0.00316x^2^ + 0.55236x − 0.06204	0.998
salicylic	y = −0.03384x^2^ + 3.26378x + 0.82676	0.998

## Data Availability

Not applicable.

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
