# Peer review of "The Effect of Fresh Kale (Brassica oleracea var. sabellica) Addition and Processing Conditions on Selected Biological, Physical, and Chemical Properties of Extruded Snack Pellets"

_molecules, 2023, doi:10.3390/molecules28041835_

Round 1

Reviewer 1 Report

An interesting research work concernig  actual issues, i.e. the possibility of fortifying snacks. Below I listed some errors that are in the manuscript, and the Authors need to be corrected:

 Introduction

line 86 - add literature

line86-89 should rewrite this sentence. Fresh kale did not have pro-healt properties, only the compounds contained in it. Which ones need to be mentioned by adding the relevant literature.

 Materials and methods

 3.1.

how long was the premix conditioned

 there is no name or type of laboratory mill

3.10.

"The content of total carbohydrates was calculated as dry matter–protein–

fat–ash–fibre" should be changed to „The content of available carbohydrates was calculated as dry matter–protein–fat-ash-fiber”.

 The results and discussions

 The rules of determination should not be given in the results and discussions. (line 130-135 is for the FRAP method. This should be removed. See the throught  manuscript and remove the determination  rules from the text.

DPPH should be reported as TEAC mgTrolox/g or kg products. The DPPH determined by  radical scavending ability (RSA) in  percentage was  useless. Look at Figure  1 DPPH reduces the active radical intensively for 10 minutes. And in this  minute convert  RSA  to TEAC.

 Line156-160 delete or move to the Introduction. The sentence in this place in the text is not necessary. At this point, I would add what new compounds formed during extrusion affect the antioxidant activities.

Correlations between DPPH and FRAP and TPC should be calculated.

Why in table3” Results of phenolic acids content in snack pellets with 30% of fresh kale” there is only a quantitative and qualitative profile for only one sample. All samples must be tested.

WSI and WAI are affected differently by dietary fiber, please raise this issue when discussing the results of these two parameters

Conclusions

Conclusions should be more compact

Author Response

The authors would like to thank the Reviewer for valuable comments which have helped to improve the quality of the manuscript. We hope that the revisions in the manuscript and accompanying responses will be sufficient to make our manuscript suitable for publication.

An interesting research work concernig  actual issues, i.e. the possibility of fortifying snacks. Below I listed some errors that are in the manuscript, and the Authors need to be corrected:

Introduction

line 86 - add literature

Thank you for valuable comment. Literature has been added.

line86-89 should rewrite this sentence. Fresh kale did not have pro-healt properties, only the compounds contained in it. Which ones need to be mentioned by adding the relevant literature.

Thank you for your accurate comment. It has been corrected.

 Materials and methods

 3.1.

how long was the premix conditioned, there is no name or type of laboratory mill

Blends of potato components with fresh kale were kept for at least 0.5 h before extrusion to uniform the moisture distribution. Laboratory grinder LMN100 (TestChem, Radlin, Poland) was used in experiment. Proper information was added to MM 3.1 section.

3.10.

"The content of total carbohydrates was calculated as dry matter–protein– fat–ash–fibre" should be changed to „The content of available carbohydrates was calculated as dry matter–protein–fat-ash-fiber”.

The sentence has been corrected.

The results and discussions

The rules of determination should not be given in the results and discussions. (line 130-135 is for the FRAP method. This should be removed. See the throught  manuscript and remove the determination  rules from the text.

Thank you for your accurate comment. The rules of  FRAP and UPLC methods have been removed from Results.

DPPH should be reported as TEAC mgTrolox/g or kg products. The DPPH determined by  radical scavending ability (RSA) in  percentage was  useless. Look at Figure  1 DPPH reduces the active radical intensively for 10 minutes. And in this  minute convert  RSA  to TEAC.

Thank you for valuable comment. The TEAC values have been provided in Table 1. The results are presented as µg Trolox/g products.

Line156-160 delete or move to the Introduction. The sentence in this place in the text is not necessary. At this point, I would add what new compounds formed during extrusion affect the antioxidant activities.

Thank you for your accurate comment. It has been corrected.

Correlations between DPPH and FRAP and TPC should be calculated.

Thank you for valuable comment. Correlation have been presented in Table 4.

Why in table 3” Results of phenolic acids content in snack pellets with 30% of fresh kale” there is only a quantitative and qualitative profile for only one sample. All samples must be tested.

Thank you for your accurate comment. Content of phenolic acids was performed for sample revealing the highest phenolics content, most advantageous chemical composition and high antioxidant properties.

These were the assumptions of the LIDER project from which the research was financed. Unfortunately, due to financial conditions and availability of equipment, we were able to determine the content of phenolic acids only for the best sample.

WSI and WAI are affected differently by dietary fiber, please raise this issue when discussing the results of these two parameters

It was added to the text:

Analyzing the results of WAI and WSI it can be generally stated that higher fiber content in enriched snacks resulted limiting of WAI due to the replacement of starch by fibrous fractions of plants. Increased fiber content was responsible for lowering WSI because of low solubility of fibrous plant fractions during the test.

Conclusions

Conclusions should be more compact

Conclusions have been shortened according to Reviewer comment.

Reviewer 2 Report

Dear Authors

I highly appreciate the research carried out by your team to determine the effect of the addition of fresh kale on the antioxidant activity and selected physicochemical properties of extruded snack pellets. The subject of the work is current and socially important due to the expectations of the current consumer, who more and more often pays attention to the composition and health-promoting properties of food. I have some comments on the manuscript:

- Lines 9-22 - please standardize the method of giving authors' affiliations

- Line 82 - please complete reference to literature (Korus et al.)

- Line 198 - please put a dot after the abbreviation (al.)

- Line 243 - please explain the meaning of the abbreviations WAI and WSI (give full name and abbreviations in brackets)

- Line 296 - Name Mismatch (References - Acosta-Perez)

- Line 383 - please change the value range for fat (0.07-0.11 g/100 g)

- Lines 384, 386, 387 - please insert spaces (before g)

- Line 442 - please change the value of PC1 (55.36%)

- Line 501 - please rewrite (2.5 mL)

- Line 555 - please change reference to reference [41]

- Line 556 - please change reference to reference [42]

Please complete References with DOI numbers

Best regards

Author Response

The authors would like to thank the Reviewer for valuable comments which have helped to improve the quality of the manuscript. We hope that the revisions in the manuscript and accompanying responses will be sufficient to make our manuscript suitable for publication.

Dear Authors

I highly appreciate the research carried out by your team to determine the effect of the addition of fresh kale on the antioxidant activity and selected physicochemical properties of extruded snack pellets. The subject of the work is current and socially important due to the expectations of the current consumer, who more and more often pays attention to the composition and health-promoting properties of food. I have some comments on the manuscript:

 - Lines 9-22 - please standardize the method of giving authors' affiliations

Thank you for valuable comment. It has been corrected.

- Line 82 - please complete reference to literature (Korus et al.)

Thank you for your accurate comment. It has been corrected

- Line 198 - please put a dot after the abbreviation (al.)

It has been corrected.

- Line 243 - please explain the meaning of the abbreviations WAI and WSI (give full name and abbreviations in brackets)

The meaning of the abbreviations has been explained.

- Line 296 - Name Mismatch (References - Acosta-Perez)

Reference has been corrected. It has been corrected.

- Line 383 - please change the value range for fat (0.07-0.11 g/100 g)

The value range has been changed.

- Lines 384, 386, 387 - please insert spaces (before g)

The spaces have been corrected.

- Line 442 - please change the value of PC1 (55.36%)

The value  has been changed

 - Line 501 - please rewrite (2.5 mL)

It has been corrected.

- Line 555 - please change reference to reference [41]

Reference has been corrected.

- Line 556 - please change reference to reference [42]

Reference has been corrected.

- Please complete References with DOI numbers

Thank you for your accurate comment. Where it was possible, DOI numbers were completed.

Reviewer 3 Report

Dear Authors,

The article entitled "Effect of fresh Brassica oleracea var. sabellica addition and processing conditions on selected biological, physical and chemical properties of extruded snack pellets" seems to be a promising work, the authors determined the effect of the addition of fresh kale on the antioxidant activity and selected physicochemical properties of extruded snack pellets. Nevertheless, current article contained several formal, linguistic, and logical deficiencies, therefore, unfortunately I suggest to reject in current form. However, I would like to list a few comments and questions towards the Authors which may help them improve the quality of current work and make it suitable for publishing, as the topic is highly interesting. 

Major concern:

L 478: Why you used 99.8 % methanol in extraction? What about other polar or non-polar solvents? Why not aqueous extract or even H2O/ Ethanol?

-The conclusion is long, summarize it and focus on the main findings, advantages, limitations, future perspectives.

- Table 1, why you wrote Gallic Acid Equivalent [μg/mL] findings in this table?? This table is related to antioxidant power no for TPC? Also, you wrote the TPC (as gallic acid) in Table 2?? Very confusing table??

-You must add the results expression details for 3.4. Ferric-Reducing Antioxidant Power (FRAP)? Trolox?

Special comments:

-          The title: add term “kale” before Brassica oleracea var. sabellica….

-          The abstract is confusing. Write the aim directly, the aim must be matched with the title. Numerical results along with the significant findings should be added. Focus on the findings, state the conclusion, future recommendations. Overall, rewrite the abstract.

-          Rewrite the keywords, what is the mean of selected biological? Instead of physical and chemical properties, you could write physicochemical properties.

-          L83. Write the product in “100 g of fresh product”?

-          L465: add the drying instrument model.

-          L 483. How you filtered the extract? Add the details.

-          Section 3.2. Preparation of Extracts: add reference.

-          IN 3.3. Free Radical Scavenging Activity – DPPH Method, why you read the absorbance every 5 min?? are you follow the original method?? What about ascorbic acid utility as a standard in this assay?? Add the equation of the calculation of DPPH (% antioxidant activity)???

-          Figure 1: define the time? Are you mean absorbance reading time? Define ms? Apply this issue throughout all figures/tables, by describing all details in the captions.

-          Ls 503 and 510. Don’t repeat model name. you mentioned in DPPH assay!

-          L 507: add more details for TPC method.

-          Add the references for 3.7. Water Absorption Index (WAI) & 3.8. Water Solubility Index (WSI)& 3.9. Fat Absorption Index.

-          In Table 2 what is the mean of symbol*

-          In Table 3, why you analysed only 30 %?

-          Table 6 and 7, define the fatty acid names.

Author Response

The authors would like to thank the Reviewer for valuable comments which have helped to improve the quality of the manuscript. We hope that the revisions in the manuscript and accompanying responses will be sufficient to make our manuscript suitable for publication.

Dear Authors,

The article entitled "Effect of fresh Brassica oleracea var. sabellica addition and processing conditions on selected biological, physical and chemical properties of extruded snack pellets" seems to be a promising work, the authors determined the effect of the addition of fresh kale on the antioxidant activity and selected physicochemical properties of extruded snack pellets. Nevertheless, current article contained several formal, linguistic, and logical deficiencies, therefore, unfortunately I suggest to reject in current form. However, I would like to list a few comments and questions towards the Authors which may help them improve the quality of current work and make it suitable for publishing, as the topic is highly interesting

Major concern:

- L 478: Why you used 99.8 % methanol in extraction? What about other polar or non-polar solvents? Why not aqueous extract or even H2O/ Ethanol?

Thank you for your accurate comment. In all studies on functional foods, we use dilutions of methanol or ethanol in water, most commonly 80 parts methanol : 20 parts water. In this case, we also used diluted methanol (80 H2O : 20 MeOH v/v) . The stated concentration of methanol is a typo, for which we sincerely apologize.

-The conclusion is long, summarize it and focus on the main findings, advantages, limitations, future perspectives.

Thank you for valuable comment. Conclusions have been shortened according to Reviewer comment.

- Table 1, why you wrote Gallic Acid Equivalent [μg/mL] findings in this table?? This table is related to antioxidant power no for TPC? Also, you wrote the TPC (as gallic acid) in Table 2?? Very confusing table?

Thank you for your accurate comment. The Table 2 (previously Table 1) has been modified. I agree that the table was illegible. The Table presents activity of the extracts towards Fe3+ reduction which is presented as FRAP units and gallic acid equivalents in order to facilitate comparison the activity with other results for kale available in other papers.

-You must add the results expression details for 3.4. Ferric-Reducing Antioxidant Power (FRAP)? Trolox?

The information has been added.

Special comments:

The title: add term “kale” before Brassica oleracea var. sabellica….

The word „kale” has been added to the title

- The abstract is confusing. Write the aim directly, the aim must be matched with the title. Numerical results along with the significant findings should be added. Focus on the findings, state the conclusion, future recommendations. Overall, rewrite the abstract.

Thank you for your accurate comment. The abstract has been completely rewritten

-  Rewrite the keywords, what is the mean of selected biological? Instead of physical and chemical properties, you could write physicochemical properties.

Thank you for your accurate comment. The keywords have been rewritten.

- L83. Write the product in “100 g of fresh product”?

Thank you for your accurate comment. The word product was used incorrectly. We replaced it with the word vegetable.

-  L465: add the drying instrument model.

It was added to the text:

laboratory shelf dryer (own design) with forced air circulation by two fans with 2 kW heating system, drying temperature was maximum 40°C.

-  L 483. How you filtered the extract? Add the details.

The methodology has been supplemented with information on filtration (section 3.2.).

-  Section 3.2. Preparation of Extracts: add reference.

Reference has been added.

IN 3.3. Free Radical Scavenging Activity – DPPH Method, why you read the absorbance every 5 min?? are you follow the original method?? What about ascorbic acid utility as a standard in this assay?? Add the equation of the calculation of DPPH (% antioxidant activity)?

Thank you for valuable suggestion. In accordance with available methods and my experience with the method, free radical scavenging activity should be observed within at least 30 minutes by every 5 minutes (or shorter time) in order to observe kinetic of the reaction. In many cases differences in kinetic can be observed within first few minutes what allow to evaluate the sample activity more detailed.  

The presented studies were based on analysis of production parameters (moisture content, screw rotation speed and kale content on pro-health properties of analyzed functional food sample thus the comparison to ascorbic acid was not performed. Nevertheless, the studies were enriched with trolox standard (Table 1.)

Figure 1: define the time? Are you mean absorbance reading time?

The time is a absorbance reading time (time from reaction initiation).

- Define mc? Apply this issue throughout all figures/tables, by describing all details in the captions.

The abbreviations used have been explained under the tables and figures

-  Ls 503 and 510. Don’t repeat model name. you mentioned in DPPH assay!

The model name has been removed.

- L 507: add more details for TPC method.

The additional information was added.

- Add the references for 3.7. Water Absorption Index (WAI) & 3.8. Water Solubility Index (WSI)& 3.9. Fat Absorption Index.

The references have been added.

-   In Table 2 what is the mean of symbol*

The asterisk was meant to refer to the abbreviations explained under the table, it was removed.

- In Table 3, why you analysed only 30 %?

Thank you for your accurate comment. Content of phenolic acids was performed for sample revealing the highest phenolics content, most advan-tageous chemical composition and high antioxidant properties.

These were the assumptions of the LIDER project from which the research was financed. Unfortunately, due to financial conditions and availability of equipment, we were able to determine the content of phenolic acids only for the best sample.

Table 6 and 7, define the fatty acid names.

The fatty acid names have been defined.

Round 2

Reviewer 1 Report

 In my opinion the Authors have corrected most of the comments.

 I am asking the Authors to remove Fig1. which is completely unnecessary (because it's not even reaction kinetics). Please leave only DPPH in terms of TEAC (after 10 min) -  table 1. Please also refer to the results of antioxidant activity determined by FRAP. ( why the big difference between the activity values ​​marked DPPH and FRAP- please give the explanation in section Results and discussion with relevant references)

Author Response

The authors would like to thank the Reviewer for valuable comments which have helped to improve the quality of the manuscript. We hope that the revisions in the manuscript and accompanying responses will be sufficient to make our manuscript suitable for publication. We have made all the changes suggested in the Reviewer's comments in the text.

 I am asking the Authors to remove Fig 1. which is completely unnecessary (because it's not even reaction kinetics). Please leave only DPPH in terms of TEAC (after 10 min) -  table 1. Please also refer to the results of antioxidant activity determined by FRAP. ( why the big difference between the activity values ​​marked DPPH and FRAP- please give the explanation in section Results and discussion with relevant references).

Thank you for the suggestions. The Figure 1 has been removed. The analysis of difference between DPPH and FRAP activity was presented in Results and discussion section along with relevant two references.

Reviewer 3 Report

Dear Authors,

Thank you for your critical revision.

Please address these few comments:

Lines 32, 33, 248: dont capitalized these terms: Water Absorption Index, Water Solubility Index, Fat Absorption Index.

in Table 8 & 9: check the color type of "α-Linolenic acid, C 18:3 n-3"

Line 509: ....aqueous methanol (80 mL methanol: 20 mL H2O v/v).

Good luck

Author Response

The authors would like to thank the Reviewer for valuable comments which have helped to improve the quality of the manuscript. We hope that the revisions in the manuscript and accompanying responses will be sufficient to make our manuscript suitable for publication. We have made all the changes suggested in the Reviewer's comments in the text.

Dear Authors,

Thank you for your critical revision.

Please address these few comments:

Lines 32, 33, 248: dont capitalized these terms: Water Absorption Index, Water Solubility Index, Fat Absorption Index.

Thank you for comment. Capital letters have been corrected.

In Table 8 & 9: check the color type of "α-Linolenic acid, C 18:3 n-3"

Color type of "α-Linolenic acid, C 18:3 n-3" has been corrected.

Line 509: ....aqueous methanol (80 mL methanol: 20 mL H2O v/v).

Thank you for comment. It has been corrected.

Good luck

Thank you.